# Monolithic lithium niobate photonic circuits for Kerr frequency comb generation and modulation

Cheng Wang [1,2], Mian Zhang [1,3], Mengjie Yu[1], Rongrong Zhu[1,4], Han Hu[1,5] & Marko Loncar[1]

Microresonator Kerr frequency combs could provide miniaturised solutions for a wide range of applications. Many of these applications however require further manipulation of the generated frequency comb signal using photonic elements with strong second-order non-linearity ($\chi^{(2)}$). To date these functionalities have largely been implemented as discrete components due to material limitations, which comes at the expense of extra system complexity and increased optical losses. Here we demonstrate the generation, filtering and electro-optic modulation of a frequency comb on a single monolithic integrated chip, using a nanophotonic lithium-niobate platform that simultaneously possesses large electro-optic ($\chi^{(2)}$) and Kerr ($\chi^{(3)}$) nonlinearities, and low optical losses. We generate broadband Kerr frequency combs using a dispersion-engineered high-$Q$ lithium-niobate microresonator, select a single comb line using an electrically programmable add-drop filter, and modulate the intensity of the selected line. Our results pave the way towards monolithic integrated frequency comb solutions for spectroscopy, data communication, ranging and quantum photonics.

[1] John A. Paulson School of Engineering and Applied Sciences, Harvard University, Cambridge, MA 02138, USA. [2] Department of Electronic Engineering & State Key Laboratory of Terahertz and Millimeter Waves, City University of Hong Kong, Kowloon, Hong Kong, China. [3] HyperLight Corporation, 501 Massachusetts Avenue, Cambridge, MA 02139, USA. [4] The Electromagnetics Academy at Zhejiang University, College of Information Science and Electronic Engineering, Zhejiang University, 310027 Hangzhou, China. [5] College of Optical Science and Engineering, Zhejiang University, 310027 Hangzhou, China. These authors contributed equally: Cheng Wang, Mian Zhang. Correspondence and requests for materials should be addressed to M.L. (email: loncar@seas. harvard.edu)

Optical frequency combs are excellent broadband coherent light sources and precise spectral rulers[1,2]. Microresonator-based Kerr frequency comb generation[3], which relies on third-order optical nonlinearity ($\chi^{(3)}$), could enable a wide range of applications including optical clocks[4], pulse shaping[5], spectroscopy[6–8], telecommunications[9], light detection and ranging (LiDAR)[10,11] and quantum information processing[12]. However, most frequency comb applications require, in addition to the comb generator, a variety of photonic components such as fast switches, modulators and/or nonlinear wavelength converters, which rely on strong second-order optical nonlinearity ($\chi^{(2)}$)[4,5,9,12]. To date these functionalities have largely been implemented as discrete off-chip components[4,5,9,12], which comes at the expense of extra system complexity and increased losses.

Microresonator Kerr combs have been realised in many material platforms, including silica ($SiO_2$)[4,6,8,11,13], silicon nitride (SiN)[5,7,9,10,12,14], silicon (Si)[15], crystalline fluorides[16], diamond[17], aluminium nitride (AlN)[18] and aluminium-gallium arsenide (AlGaAs)[19]. While most of these materials possess large $\chi^{(3)}$ nonlinearity and low optical loss required for Kerr comb generation, they usually have small or zero $\chi^{(2)}$ nonlinearity and therefore are not suitable for on-chip integration of $\chi^{(2)}$ components. Carrier-injection-based Si devices can be electrically modulated at high speeds, but exhibit much higher optical losses than their intrinsic Si counterparts[20]. (Al)GaAs possesses high $\chi^{(2)}$ nonlinearity for second harmonic generation, but much weaker electro-optic effect ($r_{41} = 1.5 \times 10^{-12}$ m V$^{-1}$)[21]. As a result, on-chip manipulation of the generated combs has been limited to slow thermal effects[22] or high-voltage electrical signals[23] to date. While heterogeneous integration of photonic chips with different functionalities has been proposed to circumvent this problem[24],

this approach requires scalable and low-loss optical links between chips, which is challenging.

Here we address the challenge of achieving $\chi^{(2)}$ functionalities by the monolithic integration of lithium-niobate (LN, LiNbO$_3$) nanophotonic waveguides, microring resonators, filters and modulators on the same chip. LN is a material that simultaneously possesses large $\chi^{(3)}$ ($1.6 \times 10^{-21}$ m$^2$ V$^{-2}$) and $\chi^{(2)}$ ($r_{33} = 3 \times 10^{-11}$ m V$^{-1}$) nonlinearities[21,25]. Specifically, the $\chi^{(3)}$ nonlinearity enables the generation of Kerr frequency combs, whereas the $\chi^{(2)}$ nonlinearity (electro-optic effect) is used to manipulate the generated comb by an external electrical field (Fig. 1). We demonstrate wide-spanning (>700 nm) Kerr comb generation, electrically programmable filtering of a single comb line with a pump rejection ratio of 47 dB, and intensity modulation of the selected line at up to 500 Mbit s$^{-1}$, all achieved on an LN photonic chip.

## Results

**Dispersion engineering and Kerr comb generation.** In order for the $\chi^{(3)}$ optical parametric oscillation (OPO) process to take place, a microresonator with a high-quality ($Q$) factor and anomalous dispersion is needed. The former ensures that the four-wave mixing process could cascade and overcome the optical losses of the microresonator, and the latter compensates for the nonlinear responses of the strong pump, i.e. self-phase modulation and cross-phase modulation[13]. While ultra-high-$Q$ (~$10^8$) LN whispering-gallery-mode resonators have been demonstrated using mechanical polishing methods[26], their dispersion properties are predetermined by the bulk material properties and cannot be engineered. In contrast, our integrated approach relies on an ultralow-loss micro-structured LN photonic platform that offers

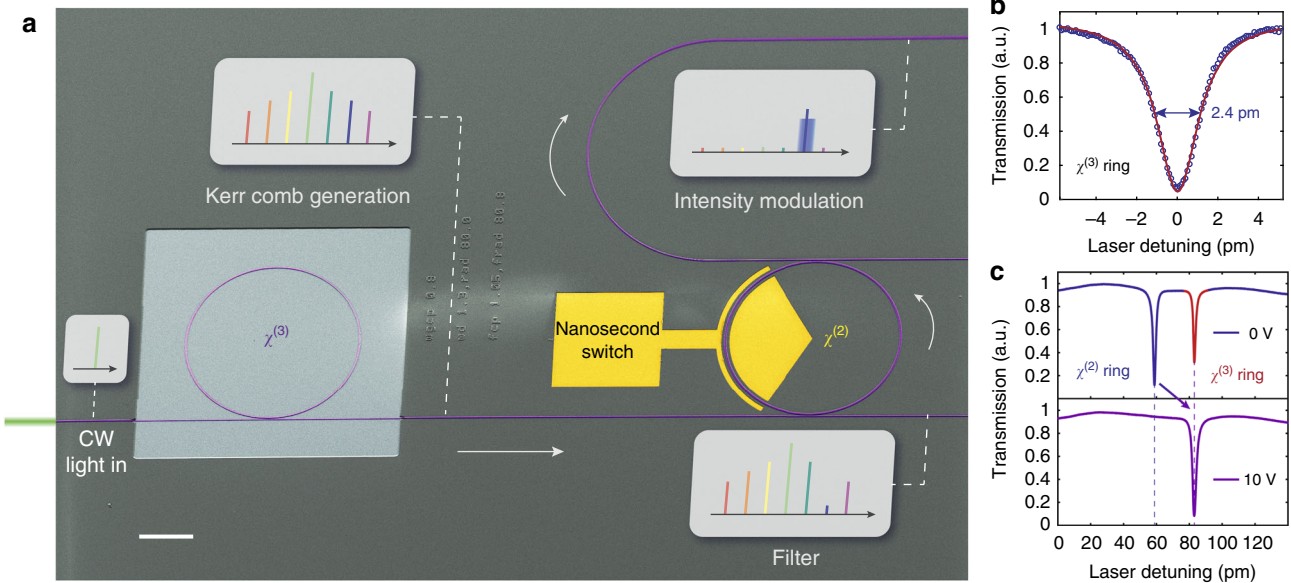

**Fig. 1** Monolithic integrated photonic circuit for frequency comb generation and manipulation. **a** A false-colour scanning electron microscope (SEM) image showing a fabricated lithium-niobate nanophotonic circuit that consists of a microresonator frequency comb generator ($\chi^{(3)}$) and an electro-optically tuneable add-drop filter ($\chi^{(2)}$). The comb generation area is air cladded to achieve anomalous dispersion, whereas the rest of the chip is cladded in SiO$_2$. Continuous-wave (CW) pump light first passes through the dispersion-engineered microring resonator to generate a frequency comb. The generated frequency comb is then filtered by an add-drop microring filter. At the drop port of the filter, a single target comb line is selected by applying an external bias voltage on the integrated electrodes to align the filter passband with the comb line. Finally, the selected comb line can be modulated at high speeds via the $\chi^{(2)}$ effect. Scale bar: 50 μm. **b** Optical transmission spectrum of the $\chi^{(3)}$ microring resonator. The measured loaded (intrinsic) quality ($Q$) factor of transverse-electric (TE) polarised mode is 6.6×10$^5$ (1.1×10$^6$). **c** Transmission spectra at the through port when different direct-current (DC) bias voltages are applied. At zero bias (top plot), the comb resonance (red dip) has a 24-pm mismatch with the filter resonance (blue dip). Applying a bias of 10 V can align the two resonances (bottom plot), showing a measured electrical tuning efficiency of 2.4 pm V$^{-1}$.

dispersion engineering capability. Our platform utilises a single-crystal LN film with sub-micron thickness bonded on top of an SiO₂ substrate[27–36]. By lithography and dry etching of the thin LN film, microresonators that have Q factors up to 10 million[27], and that allow dispersion engineering[28], can be realised. Using an x-cut LN thin-film wafer, we achieve anomalous dispersion in the telecom wavelength range for both the transverse-electric (TE) and transverse-magnetic (TM) polarisations by carefully engineering the waveguide width and thickness (Fig. 2a, b). Our dispersion-engineered microring resonator feature a loaded (intrinsic) Q factor of $6.6\times10^5$ ($1.1 \times 10^6$) for TE polarisation, as shown in Fig. 1b, resulting in an estimated OPO pump threshold of ~80 mW (Supplementary Note 1)[13]. The loaded (intrinsic) Q factor of the TM mode is $6.0\times10^5$ ($9.2\times10^5$). The measured Q factors are lower than our previous results[27] due to a reduced waveguide top width and the use of an air cladding, which are required for dispersion engineering in the current design.

For a microring resonator with a radius of 80 μm and a top width of 1.3 μm, we observe broadband frequency comb generation for both TE-like and TM-like polarisation modes at a pump power of ~300 mW in the input bus waveguide, with a comb line spacing of ~2 nm (250 GHz) (Fig. 2c, d). The measured TM-polarised comb spectrum is ~300 nm wide, while the TE-polarised comb spans from 1400 to 2100 nm, over two-thirds of an octave. The envelopes of the comb spectra, as well as relative intensity noise (RIN) measurement results (Supplementary Figure 1), indicate that the generated combs are not in a soliton state, i.e. are modulation instability frequency combs[16]. Our further investigation reveals that the spur-like features in the comb spectra could be matched to various Raman modes of LN crystal (Supplementary Figure 2)[37]. Soliton states can potentially be achieved by engineering the resonator free-spectral range (FSR) to avoid the Raman modes[38], as well as using temporal scanning techniques that have been deployed in other material platforms[16].

Importantly, our integrated LN resonators can sustain high optical powers (~50 W of circulating power), unlike their bulk/ion-diffused LN counterparts, where the photorefractive effect often causes device instability and/or irreversible damage. In our devices, the photorefractive effect shows quenching behaviour at high pump powers (>50 mW in the waveguide) (Supplementary Figure 3), similar to what was previously observed[33]. As a result, the thermal bistability effect dominates, allowing us to stably position the laser detuning with respect to cavity resonance. Despite the high circulating power inside our resonators, we do not observe optical damage after many hours of optical pumping.

**On-chip manipulation of the generated Kerr combs**. We achieve the filtering and fast modulation of comb signals by integrating an electrically tuneable add-drop filter with the comb generator on the same chip (Fig. 1). The add-drop filter consists of an LN microring resonator whose FSR is designed to be ~1% larger than the comb generator (Supplementary Figure 4). The slightly detuned FSR utilises the Vernier effect to enable the selection of a single optical spectral line over a wide optical band. The filter ring is over-coupled to both the add and the drop bus waveguides with the same coupling strength, to ensure a high extinction ratio (on/off ratio). When the input light is on (off) resonance with the filter, the majority of the optical power at the wavelength of interest will be transmitted to the drop (through) port of the filter. The current single-ring architecture results in a Lorentzian-shaped filter transfer function (Supplementary Figure 5). Advanced filter designs such as coupled-resonator optical waveguide could allow for flat-top filter responses[39]. Importantly, the microring filter is integrated with metal electrodes positioned closely to the ring. This allows for fast and efficient tuning of the filter frequency (Fig. 1c), as well as amplitude modulation of the dropped light, via the electro-optic effect. In order to access the maximum electro-optic coefficient ($r_{33}$), we design the two resonators to operate both in TE modes. The comb ring and the filter ring are cladded with air and SiO₂ respectively (Fig. 1a), to ensure that both devices operate in their best configurations (see Methods).

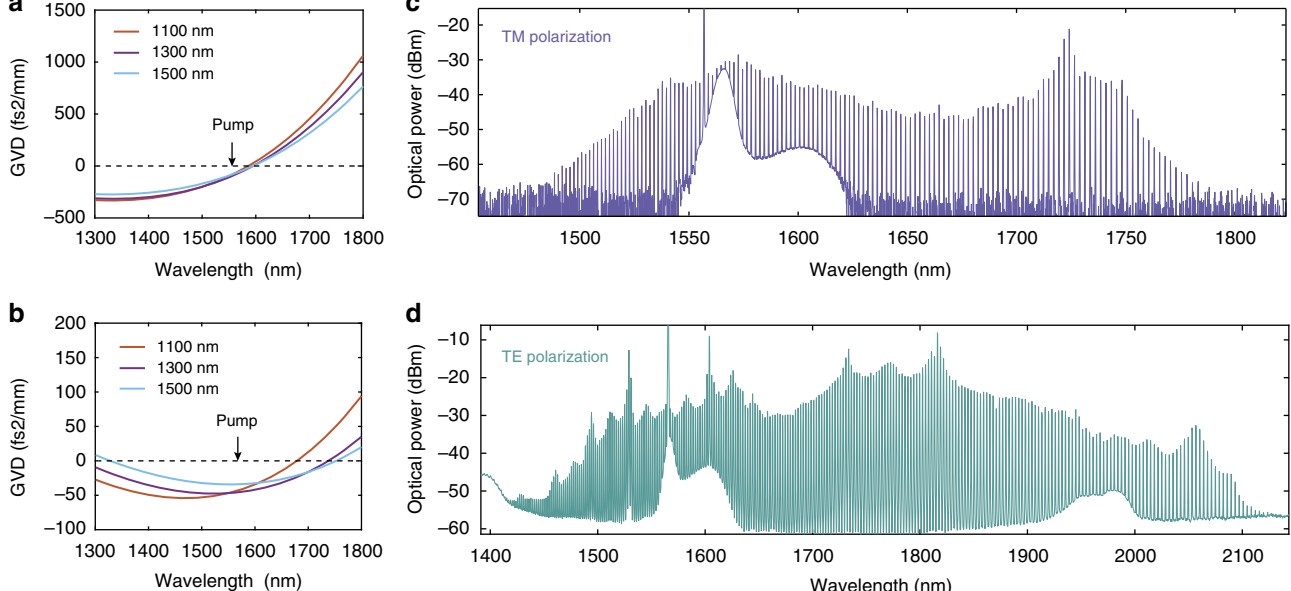

**Fig. 2** Broadband frequency comb generation. **a**, **b** Numerically simulated group-velocity dispersions (GVD) at telecom wavelengths for LN waveguides with different top widths. Anomalous dispersions (GVD < 0) can be achieved for both transverse-magnetic (TM) (**a**) and transverse-electric (TE) (**b**) modes. **c**, **d** Generated frequency comb spectra when the input laser is tuned into resonance with either TM (**c**) or TE (**d**) modes at a pump power of ~300 mW in the bus waveguide. The generated combs have a line spacing of ~ 250 GHz, and span ~300 nm (**c**) and ~700 nm (**d**) for TM and TE modes, respectively

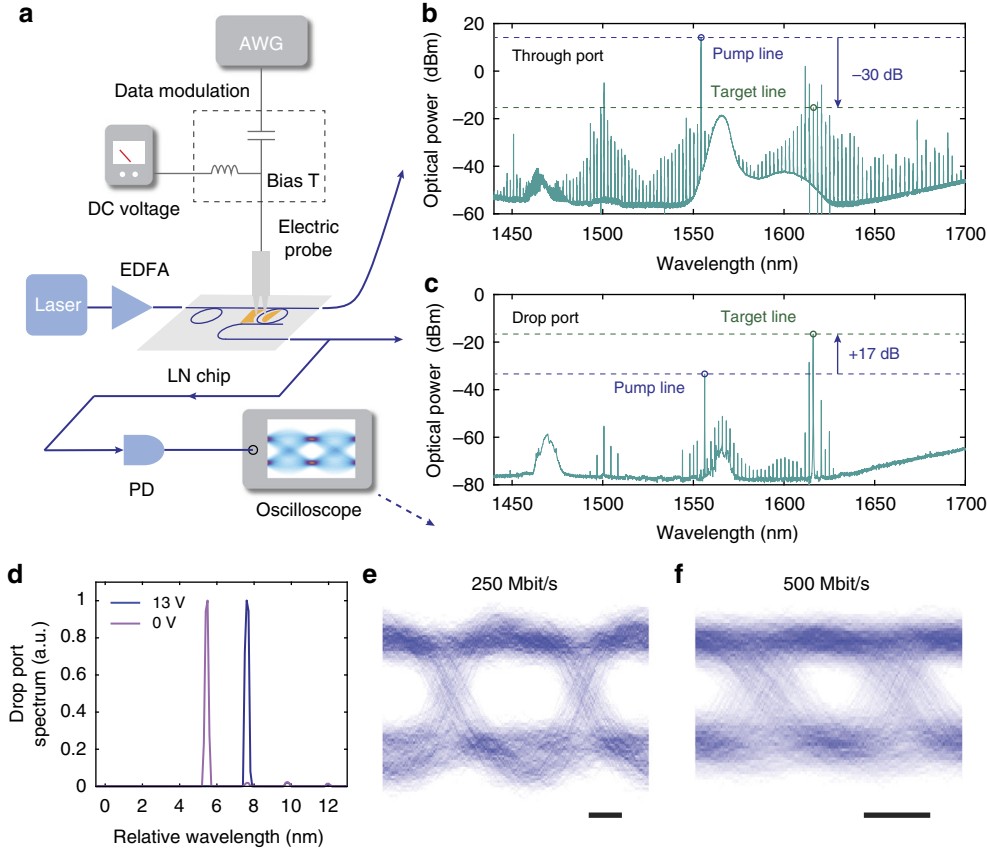

**Fig. 3** On-chip filtering and modulation of a frequency comb. **a** Simplified characterisation setup. **b**, **c** Measured optical spectra at the through (**b**) and the drop (**c**) ports of the filter, picking out a target comb line at ~1616 nm. The filter shows 47 dB suppression of the pump light. **d** Zoom-in view of the drop-port output spectra near the target line at different DC bias voltages. Applying a bias voltage of 13 V shifts the target from one comb line to the next one. **e**, **f** Applying AC electric signals could modulate the intensity of the selected comb line at 250 Mbit s$^{-1}$ (**e**) and 500 Mbit s$^{-1}$ (**f**). Eye diagrams are measured by sending a random-binary voltage sequence to the filter, and monitoring the real-time output optical power. Open-eye operations can be achieved for both bit rates. Scale bars: 1 ns. AWG arbitrary waveform generator, EDFA erbium-doped fibre amplifier, PD photodetector

We show efficient filtering of a single comb line and the strong suppression of pump light using the on-chip filter (Fig. 3a−d). We apply a direct-current (DC) bias voltage to align the filter frequency with a target comb line at 1616 nm (Fig. 1c). In this case the pump frequency at ~1556 nm has a 730-pm mismatch with the filter resonance, resulting in an experimentally measured 47 dB rejection of the pump power in the drop port (Fig. 3b, c). The filter also shows ~20 dB extinction for the comb lines adjacent to the target line (Fig. 3b, c). The measured filter extinction ratios agree well with theoretical predictions (Supplementary Figure 5). Different target comb lines can be selected by applying different bias voltages (Fig. 3d). The required additional bias voltage to change the target comb line to the adjacent one is measured to be ~13 V (Fig. 3d). This is consistent with the measured electro-optic tuning efficiency of ~2.4 pm V$^{-1}$ and the FSR difference between the comb resonator and filter resonator of 27 pm.

We show the selected target comb line at the drop port can also be modulated at speeds up to 500 Mbit s$^{-1}$, orders of magnitude faster than tuning method based on thermo-optic effects[22]. We use an arbitrary waveform generator (AWG) to deliver random-binary voltage sequences to the electrodes of the filter ring, in addition to the DC bias voltage (Fig. 3a). The peak-to-peak modulation voltage in this case is 1.5 V, sufficient to tune the filter passband (~3 pm wide) away from the target comb line. At data rates of 250 and 500 Mbit s$^{-1}$, we demonstrate open-eye data operation of the filtered comb line (Fig. 3e, f). The electro-optic

bandwidth of our filter/modulator (~400 MHz) is currently limited by the photon lifetime of the resonator (0.4 ns). The modulation speed can be dramatically improved (beyond 100 Gbit s$^{-1}$) by integrating a Mach-Zehnder modulator after the tuneable microring filter[40].

## Discussion

In summary, we have demonstrated Kerr comb generation followed by spectral and temporal manipulation of the comb signal, all achieved on the same LN chip. Our platform could lead to a new generation of photonic circuits based on the monolithic integration of frequency comb generators with both passive and active photonic components. Leveraging the giant effective $\chi^{(3)}$ nonlinearity in a quasi-phase-matched $\chi^{(2)}$ waveguide, frequency comb generation with much lower threshold power could potentially be achieved[41]. Directly embedding electro-optic modulation in the comb generator could lead to active mode locking of a Kerr frequency comb. Further integrating the frequency comb source with multiplexer/demultiplexer and ultrafast electro-optic modulators on the same chip could provide compact and low-cost dense-wavelength-division multiplexing solutions for future ultra-broadband optical fibre communication networks[9]. The fast and independent control of the amplitude and phase of each comb line are promising for chip-scale LiDAR systems[10,11], programmable pulse shaping[5] and quantum information processing[12].

## Methods

**Device design and simulation**. Waveguide dispersion diagrams and mode profiles are numerically calculated using a commercial Finite Difference Eigenmode (FDE) solver (Lumerical, Mode Solutions). Numerical simulation shows that, for the current device layer thickness of 600 nm, air cladding is necessary for anomalous dispersions. For the filter ring, however, an $SiO_2$ cladding gives rise to a better electro-optic tuning efficiency[29]. Therefore, in the final chip, the $SiO_2$ cladding in the comb generator area is intentionally removed, while the rest of the chip, including the filter ring, is cladded (Fig. 1). The filter tuning efficiency of 2.4 pm $V^{-1}$ is lower than our previous results[29] since only one arm of the ring resonator is modulated.

**Device fabrication**. Devices are fabricated from a commercial x-cut LN-on-insulator (LNOI) wafer (NANOLN) with a 600-nm device layer thickness. Electron-beam lithography (EBL, 125 keV) is used to define the patterns of optical waveguides and microring resonators in hydrogen silsesquioxane resist (FOX®-16 by Dow Corning) with a thickness of 600 nm. The resist patterns are subsequently transferred to the LN film using $Ar^+$-based reactive ion etching, with a bias power of ~112 W, an etching rate of ~30 nm $min^{-1}$ and a selectivity of ~1:1[27,30]. The etching depth is 350 nm, with a 250-nm LN slab unetched. The coupling bus waveguide has a width of ~800 nm and the coupling gap is ~800 nm. A 1.5-μm-thick PMMA EBL resist is spun coated and exposed using a second EBL step with alignment, to produce the microelectrodes of the filter ring via a lift-off process. The structures are then cladded with an 800-nm-thick $SiO_2$ layer using plasma-enhanced chemical vapour deposition. The oxide cladding in the comb generation areas is then removed through a photolithography step followed by hydrofluoric acid (HF) wet etching to realise air-cladded devices with the required anomalous dispersions. Finally, the chip edges are diced and polished to improve the fibre-chip coupling.

**Characterisation of the comb generation and modulation**. For frequency comb characterisation, continuous-wave light from a tuneable telecom laser (Santec TSL-510) is amplified using an erbium-doped fibre amplifier (EDFA, Amonics). A three-paddle fibre polarisation controller is used to control the polarisation of input light. Tapered lensed fibres are used to couple light into and out from the waveguide facets of the LN chip. The output light is sent into an optical spectrum analyser (Yokogawa) for analysis. For filter control and manipulation, TE-polarised modes are used to exploit the highest electro-optic tuning efficiency. DC signals from a voltage supply (Keithley) and AC signals from an AWG (Tektronix 70001 A) are combined using a bias-T, before being sent to the filter electrodes using a high-speed ground-signal (GS) probe (GGB Industries). The output optical signal from the drop port is sent to a 12-GHz photodetector (Newport 1544A), and analysed using a 1-GHz real-time oscilloscope (Tektronix).

## Data availability

The data sets generated and/or analysed during the current study are available from the corresponding author on reasonable request.

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

## Acknowledgements

The authors thank Y. Okawachi and C. Reimer for valuable discussions. This work is supported in part by National Science Foundation (NSF) (ECCS1609549, ECCS-1740296 E2CDA), by Harvard University Office of Technology Development (Physical Sciences

and Engineering Accelerator Award), DARPA SCOUT program (W31P4Q-15-1-0013), and City University of Hong Kong Start-up Funds. Device fabrication is performed at the Harvard University Center for Nanoscale Systems (CNS), a member of the National Nanotechnology Coordinated Infrastructure Network (NNCI), which is supported by the National Science Foundation under NSF ECCS award no. 1541959.

## Author contributions

C.W., M.Z. and M.L. conceived the experiment. C.W. and M.Z. fabricated the devices. C. W., M.Y., R.Z. and H.H. performed numerical simulations. C.W., M.Z., M.Y., and R.Z. carried out the device characterisation. C.W. wrote the manuscript with contribution from all authors. M.L. supervised the project.

## Additional information

**Competing interests:** C.W., M.Z. and M.L. are involved in developing lithium-niobate technologies at HyperLight Corporation. The other authors declare no competing interests.

