## [Peer Review File · Nature Communications]

Reviewers' comments:

Reviewer #1 (Remarks to the Author):

The authors demonstrate on-chip integration of a Kerr comb generator and a tunable filter/modulator. Utilizing both second and third order nonlinearity of lithium niobate, broadband Kerr combs are generated with reasonable pump power, and a specific comb line can be selected and modulated for data transmission. These results will be of interest to the comb community and are suitable for publication in Nature Communication. In particular, it is first time to my knowledge that the lithium niobate system has been employed at this level of integration functionality in a comb system. The following comments should be addressed:

1. Q of the LiNbO₃ resonators: The device used in this work shows an intrinsic Q factors around 1M. However the authors also reported resonators with Q exceeding 10M in ref 26. What is limiting the Q of this current device? Is it a result of dispersion engineering, i.e., reduced top width? This should be clarified in the discussion including source of Q degradation.
2. Noise of the Kerr comb: The authors mentioned that the combs are in the modulational instability regime, where the intensity and frequency of the comb lines may exhibit high noise. Kerr-comb based telecommunications requires low noise Kerr comb states [Ref. 8 and Pfeifle, Joerg, et al. "Coherent terabit communications with microresonator Kerr frequency combs." Nature photonics 8.5 (2014): 375.] The intensity noise of a typical comb line should therefore be characterized (i.e., RIN measurement) and then reported somewhere in the manuscript. The eye patterns provided in Fig. 3 are certainly important, but RIN data can help the readers assess how the results would scale to higher bandwidths than reported in this work.
3. The comb spectrum in Fig. 2d exhibits periodic spur-like features. Is this due to type II comb operation, i.e., primary comb formation on a line spacing exceeding the FSR? Please clarify. These features are absent in TM comb in Fig. 2c. Can the authors provide discussion of what mechanism is causing the difference in these spectra. The TM comb also features an enhancement (near 1725 nm) that could evolve into a dispersive wave should this device achieve soliton generation. Could the authors comment on that feature. Most of all, the TM spectrum is remarkable clean. Its appearance suggests that, with a little effort, it could be triggered for soliton generation. I would strongly urge the authors to explore that polarization for soliton generation – although this is not essential for the current paper.
4. In addition to reference 7, the following citation should be added for dual comb spectroscopy by microcombs: Myoung-Gyun Suh, et. al. "Microresonator soliton dual-comb spectroscopy" Science 354 (6312), 600-603 (2016).

Reviewer #2 (Remarks to the Author):

Manuscript#: NCOMMS-18-27241

Title: Monolithic photonic circuits for Kerr frequency comb generation, filtering and modulation

The authors, Wang et al, report the generation, filtering, and electro-optic (EO) modulation of an optical frequency comb (OFC) on a photonic chip.

There are three major components to this manuscript,

1. The demonstration of a (noisy) optical frequency comb (OFC) in a microring,
 2. The cascading of a 2nd microring for on-chip filtering, and
 3. Low-speed electro-optic (EO) modulation in the 2nd microring,
- and the manuscript stresses on the simultaneous demonstration of these three functions on a monolithic chip.

While each of #1,2, and 3 by itself is relatively trivial now (OFCs have been and are used and applied frequently, microrings have been cascaded plenty, and microring modulators and filters are commonplace), the simultaneous demonstration of #1,2 and 3 on a single chip is novel in the

context presented here. It constitutes interesting work that is a first step towards realizing the full capability of the lithium niobate on insulator (LNOI) photonic platform used by the authors here.

That being said, I have some major concerns (listed below in their order of appearance in the manuscript) that I would like to see addressed prior to supporting publication in any journal.

1. The authors refer to their previous work, Ref. 26, for their lithography and dry etching techniques. However, Ref. 26 offers no meaningful details whatsoever for this, apart from the etching bias power of 112 W, one parameter out of many. The authors should aim to provide full and transparent details of their fabrication process by adding them to this manuscript. The fabrication process used here is what arguably permits this entire work, such as the generation of the OFC to begin with. Even if their processes may transfer differently to other fabrication facilities, it is incumbent on the authors to do their very best to provide all relevant information in their manuscript.

2. A noisy OFC, even in the context of this manuscript, is underwhelming. A dissipative Kerr soliton (DKS) comb would be preferable and appropriate. DKS states are preferable for communications because of their intrinsic lower phase noise. Realizing low phase noise non-DKS combs is possible and has been shown but is also challenging. DKS combs are also now the standard for OFCs. See for example other recent related OFC papers published recently in Nature Communications: Lee et al, Nature Comm. Vol. 8, Article number: 1295 (2017) and, Karpov et al, Nature Comm. Vol. 9, Article number: 1146 (2018).

3. The authors should provide detailed data for their observed photorefractive quenching as this is key to their ability to generate a frequency comb. Furthermore, the authors should show a frequency scan of an optical resonance similar to Fig. 1b but at high power to show the thermal bistability. Doing so will be of both practical and theoretical importance.

Other additional comments:

1. Choice of references: There is room to improve on the references used here.

a) It appears that the authors haven't referenced the seminal paper on electro-optic tuning of LNOI microrings (Guarino et al, Nat. Photonics 1(7), 407-410 (2007)). They can consider adding it.

b) Similarly, there are 3 recent reviews on LNOI that the authors should consider adding to their manuscript to provide an adequate background to their work, particularly in the context of LNOI:

A. Boes, B. Corcoran, L. Chang, J. Bowers, and A. Mitchell, "Status and Potential of Lithium Niobate on Insulator (LNOI) for Photonic Integrated Circuits," Laser Photon. Rev. 12(4), 1700256 (2018)

A. Rao and S. Fathpour, "Compact lithium niobate electrooptic modulators," IEEE J. Sel. Top. Quantum Electron. 24, 1-14 (2018).

A. Rao and S. Fathpour, "Heterogeneous thin-film lithium niobate integrated photonics for electrooptics and nonlinear optics," IEEE J. Sel. Top. Quantum Electron. 2018, 24, 8200912.

c) The authors can also briefly comment on and reference some of the advanced filter and tunable filter work from the field of silicon photonics. One such example can be

Fengnian Xia, Mike Rooks, Lidija Sekaric, and Yurii Vlasov, "Ultra-compact high order ring resonator filters using submicron silicon photonic wires for on-chip optical interconnects," Opt. Express 15, 11934-11941 (2007)

d) The authors may also wish to update reference 33 to the now published journal article.

2. Line 122: "high speeds" – regardless of the context, 500 Mbit/s is not fast. It may be faster (say compared to the 1 Mbit/s shown earlier), but as an absolute, it is not fast. As the authors themselves have noted, cutting edge modulators operate at > 100 Gbit/s. Please change the language here.

3. Additional details: The authors can add details regarding the coupling waveguide width and gap, the etch depth, etc. These details are important for providing a complete picture of this work.

Reply to reviewers' comments

Review #1

The authors demonstrate on-chip integration of a Kerr comb generator and a tunable filter/modulator. Utilizing both second and third order nonlinearity of lithium niobate, broadband Kerr combs are generated with reasonable pump power, and a specific comb line can be selected and modulated for data transmission. These results will be of interest to the comb community and are suitable for publication in Nature Communication. In particular, it is first time to my knowledge that the lithium niobate system has been employed at this level of integration functionality in a comb system. The following comments should be addressed:

Response: We thank the referee for the positive feedback and pointing out the importance and novelty of our results.

Comment #1

Q of the LiNbO₃ resonators: The device used in this work shows an intrinsic Q factors around 1M. However the authors also reported resonators with Q exceeding 10M in ref 26. What is limiting the Q of this current device? Is it a result of dispersion engineering, i.e., reduced top width? This should be clarified in the discussion including source of Q degradation.

*Response: We thank the referee for pointing out the fact that Q factors here are lower than that achieved in Ref. 26. It is indeed a result of reduced waveguide top width as well as the use of air cladding structures, which are required for achieving anomalous dispersion in our current device architecture. The reason for lowered Q factors are now discussed in the **main text**.*

Comment #2

Noise of the Kerr comb: The authors mentioned that the combs are in the modulational instability regime, where the intensity and frequency of the comb lines may exhibit high noise. Kerr-comb based telecommunications requires low noise Kerr comb states [Ref. 8 and Pfeifle, Joerg, et al. "Coherent terabit communications with microresonator Kerr frequency combs." Nature photonics 8.5 (2014): 375.] The intensity noise of a typical comb line should therefore be characterized (i.e., RIN measurement) and then reported somewhere in the manuscript. The eye patterns provided in Fig. 3 are certainly important, but RIN data can help the readers assess how the results would scale to higher bandwidths than reported in this work.

*Response: We thank the referee for bringing up the relative intensity noise (RIN) of our Kerr comb generation. We have now included **additional experimental result** on the RIN measurement in the **Supplementary Figure 1**. Results confirm that the current comb generation is a high-noise (modulation instability) state.*

Comment #3

The comb spectrum in Fig. 2d exhibits periodic spur-like features. Is this due to type II comb operation, i.e., primary comb formation on a line spacing exceeding the FSR? Please clarify. These features are absent in TM comb in Fig. 2c. Can the authors provide discussion of what mechanism is causing the difference in these spectra. The TM comb also features an enhancement (near 1725 nm) that could evolve into a dispersive wave should this device achieve soliton generation. Could the authors comment on that feature. Most of all, the TM spectrum is remarkable clean. Its appearance suggests that, with a little effort,

it could be triggered for soliton generation. I would strongly urge the authors to explore that polarization for soliton generation – although this is not essential for the current paper.

Response: We thank the referee for the careful examination of the generated comb spectra. We have now performed a comprehensive study on the origin of the spur-like features in the comb spectra and concluded that they result from Raman shifts of lithium niobate crystal. The figure below shows the locations of the spur-like features in the comb spectra and their corresponding Raman modes. Each corresponding Raman mode is labelled with its symmetry and frequency shift. Both E and A mode are Raman-active optical phonon branches in LN. TO and LO correspond to transverse and longitudinal optical modes, respectively. In addition, the dispersive-wave like feature in TM-polarized device is also a result of the $A(TO)_4$ Raman mode. We have now included the corresponding discussion in the **main text**, and the figure below in the **supplementary materials**.

We also thank the referee for bringing up the possibility of achieving soliton generation in our platform. We agree that our platform is promising for soliton generation, given that the Raman effects could be suppressed by, for example engineering the FSR to avoid the Raman modes. Such engineering and the soliton generation will be subject of our future work.

Comment #4

In addition to reference 7, the following citation should be added for dual comb spectroscopy by microcombs: Myoung-Gyun Suh, et. al. "Microresonator soliton dual-comb spectroscopy" Science 354 (6312), 600-603 (2016).

Response: We thank the referee for suggesting the additional reference and has added the reference accordingly.

Review #2

There are three major components to this manuscript,

1. The demonstration of a (noisy) optical frequency comb (OFC) in a microring,
2. The cascading of a 2nd microring for on-chip filtering, and
3. Low-speed electro-optic (EO) modulation in the 2nd microring,

and the manuscript stresses on the simultaneous demonstration of these three functions on a monolithic chip.

While each of #1,2, and 3 by itself is relatively trivial now (OFCs have been and are used and applied frequently, microrings have been cascaded plenty, and microring modulators and filters are commonplace), the simultaneous demonstration of #1,2 and 3 on a single chip is novel in the context presented here. It constitutes interesting work that is a first step towards realizing the full capability of the lithium niobate on insulator (LNOI) photonic platform used by the authors here.

***Response:** We thank the referee for pointing out the novelty of our current work - achieving three important photonic functionalities simultaneously on a monolithic photonic chip. As far as we understand, achieving a frequency comb in LN is not a trivial task for the community, since it requires simultaneously achieving high optical confinement, low loss, dispersion engineering and high optical power handling in LN. To our knowledge, this is the first time that a Kerr frequency comb is reported in a nanophotonic LN platform. We also agree with the referee that although Kerr combs have been generated in many other material platforms, it is the electro-optic property of LN that allows for the demonstration of active filtering and modulation on the same chip. Therefore we believe that all three tasks achieved in our current work are coherent and they together represent a major milestone for integrated photonics.*

That being said, I have some major concerns (listed below in their order of appearance in the manuscript) that I would like to see addressed prior to supporting publication in any journal.

Comment #1

The authors refer to their previous work, Ref. 26, for their lithography and dry etching techniques. However, Ref. 26 offers no meaningful details whatsoever for this, apart from the etching bias power of 112 W, one parameter out of many. The authors should aim to provide full and transparent details of their fabrication process by adding them to this manuscript. The fabrication process used here is what arguably permits this entire work, such as the generation of the OFC to begin with. Even if their processes may transfer differently to other fabrication facilities, it is incumbent on the authors to do their very best to provide all relevant information in their manuscript.

***Response:** We thank the referee for pointing out the importance of device fabrication which enabled our current work. While we pointed our fabrication methods in the prior manuscript to other references, we have now included a detailed description of the lithography and dry etching processes, including EBL acceleration voltage, resist thickness, selectivity, etching rate and etching depth in the revised manuscript (**Methods**). We have also provided reference to our earlier work (C. Wang et al. *Opt. Express* 2014), where the fabrication flow is discussed in more details than Ref. 26. We would like to mention though, since our etching process only involves physical etching with Ar gas, the common parameters required for etching other materials, such as gas ratios, are not applicable here. For physical etching, the etching bias power is the most important parameter, which determines the energy of ions that bombard the LN substrate. Therefore, we believe that we have been as transparent as we can, and as others (e.g. excellent prior work on integrated LN photonics: A. Guarino et al. *Nat. Photonics* 2007, A. Rao et al. *Opt. Lett.* 2016, WC Jiang et al. *Sci. Rep.* 2016, SY Siew et al. *IEEE Photonics Technol. Lett.* 2016), in providing the details of our fabrication processes.*

Comment #2

A noisy OFC, even in the context of this manuscript, is underwhelming. A dissipative Kerr soliton (DKS) comb would be preferable and appropriate. DKS states are preferable for communications because of their intrinsic lower phase noise. Realizing low phase noise non-DKS combs is possible and has been shown but is also challenging. DKS combs are also now the standard for OFCs. See for example other recent related OFC papers published recently in Nature Communications:

Lee et al, Nature Comm. Vol. 8, Article number: 1295 (2017) and,

Karpov et al, Nature Comm. Vol. 9, Article number: 1146 (2018).

Response: *We agree with referee that achieving a soliton state in our platform would be preferable. However, as we discussed above, and also as referee #1 mentioned, achieving a soliton state is not the focus and falls beyond the scope of this manuscript. Our current results focus on the integration of Kerr comb generator, active filter and electro-optic modulator on a monolithic photonic chip, a first step towards fully integrated comb systems that could enable numerous applications. Although soliton generation has been realized in many different material platforms, the material properties in those platforms do not allow the additional functionalities as we show in this work. Therefore we believe that achieving a Kerr comb in LN nanophotonic platform for the first time, by itself is already a major milestone, and could generate a lot of interests and resonance in both the LN community and the frequency comb community.*

*We do agree that noise figures are essential parameters for Kerr frequency combs. Therefore, we have now included additional experimental data on the relative intensity noise (RIN) measurements of our generated comb lines. The results are shown in **Supplementary Figure 1**.*

*In addition, we have also done a detailed study on the generated frequency comb envelope, and concluded that the spur-like features in the spectra result from Raman shifts of lithium niobate crystal. We have added a new figure in the supplementary materials (**Supplementary Figure 2**), which shows the locations of the spur-like features in the comb spectra and their corresponding Raman modes. In future work, such Raman effects could potentially be suppressed by engineering the resonator FSR to avoid the Raman modes, therefore to facilitate the generation of soliton states. We have now included the corresponding discussion in the **main text**.*

Comment #3

The authors should provide detailed data for their observed photorefractive quenching as this is key to their ability to generate a frequency comb. Furthermore, the authors should show a frequency scan of an optical resonance similar to Fig. 1b but at high power to show the thermal bistability. Doing so will be of both practical and theoretical importance.

Response: *We agree with the referee that detailed data on the observed photorefractive phenomena are important for the completeness of our manuscript, and have now added the measured data as **Supplementary Figure 3**. We show the frequency scan in both forward and backward directions at three different power levels. At low optical powers, the frequency scan shows photorefractive instability (oscillatory) behaviors, making it difficult to stabilize the laser within cavity resonance. At increased optical powers, the photorefractive oscillation period becomes longer and eventually thermal bistability dominates (**Supplementary Figure 3c**), allowing us to stably park the laser inside the cavity resonance and achieve stable Kerr comb generation.*

Additional Comment #1

Choice of references: There is room to improve on the references used here.

a) It appears that the authors haven't referenced the seminal paper on electro-optic tuning of LNOI microrings (Guarino et al, Nat. Photonics 1(7), 407–410 (2007)). They can consider adding it.

Response: *Per the referee's suggestion, we have now added this reference to make the reference list more inclusive.*

b) Similarly, there are 3 recent reviews on LNOI that the authors should consider adding to their manuscript to provide an adequate background to their work, particularly in the context of LNOI:

A. Boes, B. Corcoran, L. Chang, J. Bowers, and A. Mitchell, "Status and Potential of Lithium Niobate on Insulator (LNOI) for Photonic Integrated Circuits," *Laser Photon. Rev.* 12(4), 1700256 (2018)

A. Rao and S. Fathpour, "Compact lithium niobate electrooptic modulators," *IEEE J. Sel. Top. Quantum Electron.* 24, 1–14 (2018).

A. Rao and S. Fathpour, "Heterogeneous thin-film lithium niobate integrated photonics for electrooptics and nonlinear optics," *IEEE J. Sel. Top. Quantum Elect.* 2018, 24, 8200912.

Response: *Per the referee's suggestion, we have now added two more references, one from each group, to the revised manuscript.*

c) The authors can also briefly comment on and reference some of the advanced filter and tunable filter work from the field of silicon photonics. One such example can be

Fengnian Xia, Mike Rooks, Lidija Sekaric, and Yurii Vlasov, "Ultra-compact high order ring resonator filters using submicron silicon photonic wires for on-chip optical interconnects," *Opt. Express* 15, 11934-11941 (2007)

Response: *Per the referee's suggestion, we have now added discussion on the filter response and potential use of CROW filters in the **main text**, and added the suggested paper to the reference list.*

d) The authors may also wish to update reference 33 to the now published journal article.

Response: *We thank the referee for the careful examination of our reference list and have updated the corresponding reference (now Ref. 40).*

Additional Comment #2

Line 122: "high speeds" – regardless of the context, 500 Mbit/s is not fast. It may be faster (say compared to the 1 Mbit/s shown earlier), but as an absolute, it is not fast. As the authors themselves have noted, cutting edge modulators operate at > 100 Gbit/s. Please change the language here.

Response: *Per the referee's suggestion, we have now removed the wording of "high speeds" in line 122 and used the actual modulation speed to avoid confusion. The new text reads "... can be modulated at speeds up to 500 Mbit/s, ..."*

Additional Comment #3

The authors can add details regarding the coupling waveguide width and gap, the etch depth, etc. These details are important for providing a complete picture of this work.

Response: *Per the referee's suggestion, we have now included the coupling waveguide width and gap, and the etching depth in the **Methods** section.*

List of Modifications

* All new text has been highlighted in YELLOW in the revised manuscript.

1. In Page 5, lines 11-13: New text has been added to discuss the reasons for lowered Q factors compared with previous results. The new text reads “*The measured Q factors are lower than our previous results due to a reduced waveguide top width and the use of air cladding, which are required for dispersion engineering in the current design.*”
2. In Page 5, line 19-20: New text has been added to point out that RIN measurements are included in the Supplementary Materials. The new text reads “*The envelopes of the comb spectra, as well as relative intensity noise (RIN) measurement results (Supplementary Figure 1), indicate that the generated combs are not in a soliton state, i.e. are modulation instability frequency combs.*”
3. In Page 5, line 21-24: New text has been added to discuss the origin of the spur-like features in the comb spectra and possible ways to achieve soliton states. The new text reads “*Our further investigation reveals that the spur-like features in the comb spectra could be matched to various Raman modes of LN crystal (Supplementary Figure 2). Soliton states can potentially be achieved by engineering the resonator FSR to avoid the Raman modes*”
4. In Page 6, line 22-24: New text has been added to discuss the spectral shapes of the microring filters. The new text reads “*The current single-ring architecture results in a Lorentzian-shaped filter transfer function (Supplementary Figure 5). Advanced filter designs such as coupled-resonator optical waveguide (CROW) could allow for flat-top filter responses.*”
5. In Page 7, line 19-20: The wording of “modulated at high speeds” has been changed to the actual modulation speed to avoid confusion. The new text reads “*We show the selected target comb line at the drop port can also be modulated at speeds up to 500 Mbit/s.*”
6. In Methods, Page 9, line 12-19: New text has been added to include the full device fabrication processes and detailed parameters of the ring resonators and bus waveguides. The new text reads “*Electron-beam lithography (EBL, 125 keV) is used to define the patterns of optical waveguides and microring resonators in hydrogen silsesquioxane (HSQ) resist (FOX®-16 by Dow Corning) with a thickness of 600 nm. The resist patterns are subsequently transferred to the LN film using Ar⁺-based reactive ion etching (RIE), with a bias power of ~ 112 W, an etching rate of ~ 30 nm/min and a selectivity of ~ 1:1. The etching depth is 350 nm, with a 250-nm LN slab unetched. The coupling bus waveguide has a width of ~ 800 nm and the coupling gap is ~ 800 nm.*”
7. New references (Now Refs. 8, 34, 35, 36, 37, 38, 39) have been included in the revised manuscript to improve the inclusiveness of the reference list.

8. In Supplementary Materials, measurement results of the relative intensity noise (RIN) spectra of the Kerr combs have now been included as Supplementary Note 2 and Supplementary Figure 1.
9. In Supplementary Materials, detailed study on the spur-like features in the comb spectra and their correspondence to certain Raman modes is now provided as Supplementary Note 3 and Supplementary Figure 2.
10. In Supplementary Materials, experimental data on the quenching behavior of photorefractive instability are provided as Supplementary Note 4 and Supplementary Figure 3.

REVIEWERS' COMMENTS:

Reviewer #1 (Remarks to the Author):

The authors have fully addressed all my comments and the manuscript can be published without any delay.

Reviewer #2 (Remarks to the Author):

The authors have revised their manuscript to address my previous concerns.

I would still very much have preferred to see a soliton comb but I will not insist on it, given the integration aspect of this work.

Overall, the revisions are satisfactory, and the authors' revised manuscript is suitable for publication.

Reply to reviewers' comments

Review #1

The authors have fully addressed all my comments and the manuscript can be published without any delay.

Response: *We thank the referee for the positive feedback.*

Review #2

The authors have revised their manuscript to address my previous concerns.

I would still very much have preferred to see a soliton comb but I will not insist on it, given the integration aspect of this work.

Overall, the revisions are satisfactory, and the authors' revised manuscript is suitable for publication.

Response: *We thank the referee for the positive feedback.*